# Low Mutational Burden of Extranodal Marginal Zone Lymphoma of Mucosa-Associated Lymphoid Tissue in Patients with Primary Sjogren’s Syndrome

**DOI:** 10.3390/cancers14041010

**Published:** 2022-02-17

**Authors:** Johanna A. A. Bult, Jessica R. Plaça, Erlin A. Haacke, M. Martijn Terpstra, Gwenny M. Verstappen, Frederik K. L. Spijkervet, Frans G. M. Kroese, Wouter J. Plattel, Joost S. P. Vermaat, Hendrika Bootsma, Bert van der Vegt, Arjan Diepstra, Anke van den Berg, Klaas Kok, Marcel Nijland

**Affiliations:** 1Department of Hematology, University Medical Center Groningen, 9713 GZ Groningen, The Netherlands; j.a.a.bult@umcg.nl (J.A.A.B.); w.j.plattel@umcg.nl (W.J.P.); 2Department of Pathology and Medical Biology, University Medical Center Groningen, 9713 GZ Groningen, The Netherlands; jessicaplaca@gmail.com (J.R.P.); b.van.der.vegt@umcg.nl (B.v.d.V.); a.diepstra@umcg.nl (A.D.); a.van.den.berg01@umcg.nl (A.v.d.B.); 3Pathologie Friesland, 8917 EN Leeuwarden, The Netherlands; erlin.haacke@pathologiefriesland.nl; 4Department of Genetics, University Medical Center Groningen, 9713 AV Groningen, The Netherlands; m.m.terpstra.cluster@gmail.com (M.M.T.); k.kok@umcg.nl (K.K.); 5Department of Rheumatology and Clinical Immunology, University Medical Center Groningen, 9713 GZ Groningen, The Netherlands; g.m.p.j.verstappen@umcg.nl (G.M.V.); f.g.m.kroese@umcg.nl (F.G.M.K.); h.bootsma@umcg.nl (H.B.); 6Department of Oral and Maxillofacial Surgery, University Medical Center Groningen, 9713 GZ Groningen, The Netherlands; f.k.l.spijkervet@umcg.nl; 7Department of Hematology, Leiden University Medical Center, 2300 RC Leiden, The Netherlands; j.s.p.vermaat@lumc.nl

**Keywords:** extranodal marginal lymphoma of mucosa-associated lymphoid tissue, Sjogren’s syndrome, whole-exome sequencing, mutational analysis

## Abstract

**Simple Summary:**

Patients with primary Sjogren’s syndrome (pSS) are at risk of developing extranodal marginal zone lymphoma (ENMZL) of the mucosa-associated lymphoid tissue (MALT) in the parotid glands. The genetic mechanism underlying development of MALT lymphoma in the context of pSS is unknown. The aim of our study was to define the genomic landscape of pSS-associated MALT lymphoma. For 17 localized pSS-associated MALT lymphomas, we analyzed the presence of nonsynonymous mutations, copy number alterations (CNAs) and *MALT1* translocations. pSS-associated MALT lymphomas were characterized by a low mutational load (median number of nonsynonymous somatic variants per case was 7, range 2–78) and a limited number of CNAs. Unlike the recurrent genomic aberrations observed in MALT lymphoma, which were not associated with pSS, pSS-associated MALT lacked a clear lymphoma-related profile. The data suggest that localized pSS-associated MALT lymphomas are a distinct type of ENMZL, which are genomically stable and most likely depend on a stimulatory micro-environment.

**Abstract:**

Patients with primary Sjogren’s syndrome (pSS) are at risk of developing extranodal marginal zone lymphoma (ENMZL) of the mucosa-associated lymphoid tissue (MALT) in the parotid glands. Unlike recurrent genomic aberrations observed in MALT lymphoma, which were not associated with pSS (non-pSS), it is unknown which somatic aberrations underlie the development of pSS-associated MALT lymphomas. Whole-exome sequencing was performed on 17 pSS-associated MALT lymphomas. In total, 222 nonsynonymous somatic variants affecting 182 genes were identified across the 17 cases. The median number of variants was seven (range 2–78), including three cases with a relatively high mutational load (≥24/case). Out of 16 recurrently mutated genes, *ID3*, *TBL1XR1*, *PAX5*, *IGLL5* and *APC* are known to be associated with lymphomagenesis. A total of 18 copy number alterations were detected in eight cases. *MALT1* translocations were not detected. With respect to outcome, only two cases relapsed outside of the salivary glands. Both had a high mutational load, suggesting a more advanced stage of lymphoma. The low mutational load and lack of a clear lymphoma-related mutation profile suggests that localized pSS-associated MALT lymphomas are genomically more stable than non-pSS MALT lymphomas and most likely depend on a stimulatory micro-environment.

## 1. Introduction

Primary Sjogren’s syndrome (pSS) is a chronic multisystem autoimmune disease with an estimated prevalence of 0.02–0.25% that predominantly affects adult women [1]. pSS is characterized by unspecified chronic inflammation of the exocrine glands that results in diminished gland function [2]. In addition, patients can develop extra-glandular features such as arthritis, neuropathy and renal or pulmonary involvement.

Although the initial event leading to the inflammatory response in pSS has not fully been elucidated, T-cell dependent B-cell hyperactivity gives rise to autoantibodies and increased pro-inflammatory cytokine levels [3]. Antibodies against the ribonucleoprotein antigens anti-Ro/SSA and/or anti-La/SSB can be found in ~70% of pSS patients [2]. Rheumatoid factor (RF) is present in approximately half of the patients. In some patients, RF forms insoluble aggregates when cooled below normal body temperature, leading to cryoglobulinemia [4].

The presence of cryoglobulinemia, together with low C4 levels and salivary gland enlargement, are important risk factors for lymphoma development in pSS [5]. Nearly 5–10% of pSS patients develop non-Hodgkin’s lymphoma, of which extranodal marginal zone lymphoma (ENMZL) of the mucosa-associated lymphoid tissue (MALT) is the most common [6,7]. MALT lymphoma initially starts as a polyclonal B-cell proliferation in pSS salivary glands that, over time, develops into an oligoclonal and eventually monoclonal B-cell expansion [5,8,9]. The majority of salivary gland pSS-associated MALT lymphomas express stereotypic B-cell antigen receptors that include an immunoglobulin heavy-chain variable gene region with high affinity for RF [10]. It has been proposed that sustained (auto) antigen stimulation and inflammation in pSS results in the formation of germinal centers and the accumulation of genomic aberrations in B-cells due to somatic hypermutation and class-switch recombination [3,11]. Over time, this leads to a monoclonal expansion of intra-epithelial B-cells characterized by the expression of Fc receptor-like protein 4 [9,12]. These Fc receptor-like protein-4-expressing B-cells have an aberrant transcriptome that may put these cells at high risk for malignant transformation [13].

The genetic mechanism underlying the development of pSS-associated MALT lymphomas is largely unknown. In MALT lymphoma not associated with pSS (non-pSS), recurrent chromosomal translocations involving the *MALT1* gene have been frequently observed [11,14,15]. In contrast, the majority of pSS-associated MALT lymphomas lack recurrent chromosomal aberrations [16]. Analyses of the mutational landscape of splenic marginal zone lymphoma (SMZL) and non-pSS MALT lymphoma have identified recurrently mutated genes involved in chromatin remodeling/transcriptional regulation, *NOTCH* and *NF-κB* pathways [17,18,19,20,21,22,23,24]. The mutational profile of non-pSS MALT lymphomas showed distinct profiles associated with the anatomical sites of presentation. For example, non-pSS MALT lymphomas of the salivary gland were characterized by frequent mutations in *TBL1XR1*, *GPR34* and *NOTCH2* [15,24]. However, given that previously conducted studies have focused on non-pSS MALT lymphomas, mutational analyses of pSS-associated MALT lymphomas are currently lacking. In a small series of four pSS patients with cryoglobulinemia but without MALT lymphoma, pathogenic mutations in small fractions of circulating B-cells were observed in genes that are often associated with common B-cell lymphomas, such as *CARD11*, *TNFAIP3*, *CCND3*, *ID3*, *BTG2* and *KLHL6* [4].

The aim of the current study was to define the genomic landscape of pSS-associated MALT lymphoma of the parotid gland. The identification of specific recurrent mutations or copy number aberrations and subsequent elucidation of the affected pathways will help in understanding the mechanisms by which B-cells in pSS undergo malignant transformation.

## 2. Materials and Methods

### 2.1. Patient Materials

Fifteen fresh-frozen and three formalin-fixed paraffin-embedded (FFPE) MALT lymphomas were collected from the parotid glands of 18 pSS patients at the University Medical Center of Groningen (UMCG). Patients were identified through the electronic database from the UMCG. All specimens were collected as part of standard clinical care in patients with symptomatic disease. Matched blood samples obtained for 12 of these patients were used as germline controls. The use of tissue and clinical data from the included patients was approved by the Medical Ethical Committee of the UMCG. All patients gave written informed consent. Clonality analysis using *IgH*-PCR (BIOMED-2) was performed, to confirm the presence of a clonal B-cell process, for all samples on the diagnostic FFPE material as part of standard care [25]. A review of the samples and estimation of the tumor cell percentages was carried out by experienced pathologists (AD, EH and BV) based on morphology, immunohistochemistry for CD3, CD20, CD79a, immunoglobulin light chains, IgA, IgG and IgM, and B-cell clonality analysis.

### 2.2. Fluorescence In Situ Hybridization

Fluorescence in situ hybridization (FISH) to identify *MALT1* translocations was performed using *MALT1* break-apart probes, according to the manufacturer’s instructions (Vysis-LSI-*MALT1*-Break-Apart-FISH-Probe-Kit, Abbott, Chicago, IL, USA).

### 2.3. Whole-Exome Sequencing

DNA isolation from the fresh-frozen tumor samples and white blood cell samples was performed using a standard salt and chloroform-based procedure. Isolation of DNA from FFPE tissue sections was performed with the COBAS kit, following the manufacturer’s protocol (Cobas DNA Sample Preparation kit: Roche Molecular systems, Branchburg, NJ, USA). Whole-exome sequencing was carried out using the Human Core Exome Enrichment assay according to the manufacturer’s procedures (Twist Bioscience, South San Francisco, CA, USA), using a DNA input of 100 ng per sample. Pools containing 16 individual samples were paired-end sequenced (2 × 150 nt) on a NextSeq 500 run (Illumina, San Diego, CA, USA). Raw sequencing data were de-multiplexed, trimmed and analyzed using a previously published in-house bioinformatics pipeline [26].

### 2.4. Bioinformatics Analysis

In brief, reads were aligned to the human reference genome (GRCh37/hg19) using the Burrows–Wheeler Alignment (BWA) tool version 0.7.17-r1188. The BAM files produced by BWA were pre-processed according to the GATK best practices workflow using the HaplotypeCaller over the batches [27] and a MOLGENIS compute as a workflow tool [28]. Next, base quality score was recalibrated. Variants were called using HaplotypeCaller and FreeBayes [27,29]. Variants were annotated with 1000 genome phase 3 [30], Cosmic v72 [31], the Exome Aggregation Consortium (ExAC) 0.3 databases [32] and SnpEFF 3.5 [33]. For one fresh-frozen tumor sample, whole-exome sequencing resulted in a mean target coverage of only 7 reads, and this sample was excluded from further analysis.

### 2.5. Variant Filtering

Variants were filtered using the following quality exclusion criteria: QUAL (Phred scaled probability score) < 20, Fisher Strand bias > 60, Mapping Quality (MQ) < 40, MQRankSum < −12.5, (QUAL /Depth (QD) < 2), QD/allele frequency (AF) < 8.0 and ReadPosRandSum < −20. Next, synonymous variants, variants with a frequency >2% in the 1000 Genomes Project and variants with a minor allele frequency ≥0.1% in the ExAC and/or GnomAD databases were excluded [32]. Variants with a total read count <20 and variants with a minor allele count of <4 in the tumor sample were considered inconclusive and excluded from further analysis.

Matched blood samples were used to exclude non-harmful personal variants when the minor allele count in the blood sample was >3. The results of the two variant callers were combined into one list, keeping the HaplotypeCaller read counts when the variant was called by both algorithms [27,29]. Variants called as multi-nucleotide polymorphisms by FreeBayes were actually combinations of multiple single-nucleotide variants (SNVs) (and even InDels in a few samples). These multi-nucleotide polymorphisms were split into their individual variants based on visual inspection of the aligned reads in the Integrative Genomics Viewer [34]. The individual variants were subsequently filtered as described above, using read counts indicated in the Integrative Genomics Viewer and information obtained from UCSC genome browser GRCh37/hg19 [35].

Variants in samples without matched normal blood samples and with a total read count ≥20 and a minor allele count of >3 were filtered in a similar way, including the step where multi-nucleotide polymorphisms were resolved into their individual variants.

### 2.6. Classification of Major and Minor Clonal Variants

Variants that passed all filtering steps were classified into major and minor clonal variants according to a previously published protocol [26]. In brief, for each tumor sample, the variant with the highest variant allele frequency (VAF) was determined. All other variants with a VAF ≥50% of the highest VAF in that sample were regarded as major clonal variants and considered to be present in the majority of tumor cells. Variants with a VAF <50% of the highest VAF were regarded as minor clonal variants and assumed to be present in a subset of tumor cells. Hence, minor clonal variants are unlikely to be driver mutations. For samples with a low total read count for the variant with the highest VAF (<25), the mean of the two variants with the highest VAFs was used to determine the cutoff for major and minor variants.

### 2.7. Functional Annotation and Additional Filtering Steps

A web-based version of the Open Custom Ranked Analysis of Variants Toolkit (OpenCRAVAT) was used to link somatic variants to cancer development [36]. A custom R script was applied to convert the Excel files containing the variant information into a Variant Call Format file and to enter the data into OpenCRAVAT as a tab-delimited text file. The OpenCRAVAT modules COSMIC database (module version 91.0.0, data source version v91), Cancer Gene census (module version 85.0.12, data source version 2019-03-23) and Cancer Gene Landscape (module version 1.0.9, data source version v2013.03.29) were used to annotate the filtered variant lists.

Phred-scaled Combined Annotation Dependent Depletion scores were used as an indication of the variant deleteriousness of SNVs and InDels [37]. A Combined Annotation Dependent Depletion cut-off value of 15 was used to identify putatively pathogenic variants. Non-pathogenic SNVs and InDels were removed.

Personal variants could not be removed for tumor samples without a matched blood sample, which resulted in a higher number of variants. To prevent the inclusion of non-somatic variants, we focused on variants in genes: (1) carrying somatic variants in at least one of the samples for which a control DNA sample was available, or (2) annotated by OpenCRAVAT as a tumor-suppressor gene or an oncogene in another type of cancer [36].

### 2.8. Copy Number Alterations

Copy number alterations (CNAs) were assessed with the VARSCAN software package using the on-target reads [38]. VARSCAN infers somatic CNAs using data from matched tumor-blood pairs. For the unmatched fresh-frozen tumor samples, the analysis was carried out using sequence data from an unmatched blood sample that was processed in the same pool.

## 3. Results

### 3.1. Patient Characteristics

Patient characteristics are presented in Appendix A. Almost all patients (16 out of 17) were female. The median age was 62 years (range 40–84 years). In 16 out of 17 patients, there was a limited stage I/II of MALT disease according to the Ann Arbor classification. In one out of 17 patients, there was an advanced stage IV of MALT disease. Cryoglobulinemia was present in 13 out of 17 patients (Appendix A). The presence of a monoclonal B-cell population, indicated by clear clonal peaks, was confirmed by *IgH* PCR (BIOMED-2) for all patients. Translocations involving the *MALT1* gene were not detected in any of the cases.

### 3.2. Quality Control

The median target coverage was 130x, ranging from 84 to 163 (Appendix A), and >90% of the target region had a coverage >50x.

### 3.3. Somatic Variants in Cases with a Matched Blood Sample

In total, 163 nonsynonymous somatic variants (median number of variants = 7, range 2–78) were identified in 138 genes among 12 cases with a matched blood sample (Figure 1, Appendix A). The distribution of these variants was skewed, with 10/12 case having ≤10 variants, compared to two cases where >30 somatic variants were detected (Figure 1). The VAF ranged between 2.3% and 54.2%. The nonsynonymous somatic variants comprised 113 missense variants (69%), 19 frameshifts (12%), 10 disruptive- and in-frame insertions/deletions (6%), 12 splice variants (7%), 7 nonsense variants (4%) and 2 variants disrupting the starting codon (1%) (Appendix A).

### 3.4. Somatic Variants in Cases without a Matched Blood Sample

In total, 59 nonsynonymous somatic variants (median number of variants = 11, range 3–24) were identified in 53 genes of the five cases without a matched blood sample. The distribution of variants was skewed, with <15 variants detected in four cases and 24 variants detected in the fifth case (Figure 1). The VAF ranged between 4.9% and 50.0%. The nonsynonymous somatic variants comprised 50 missense variants (85%), 5 frameshifts (8%), 1 inframe deletion (2%) and 3 splice variants (5%) (Appendix A). In the 12 cases with matched control samples, 9 of the 138 genes harboring somatic variants harbored one or more variants in at least one of the five cases without a matched control.

### 3.5. Recurrently Mutated Genes

In total, 16 recurrently mutated (>1 case) genes were identified (Figure 2, Appendix A). Five genes are involved in epithelial surface and/or extracellular matrix (*MAMDC4*, *COL14A1*, *CAMSAP3*, *TMEM2* and *MUC4*). Five genes, all with variants in two cases, were shown to be associated with lymphoma pathogenesis (*ID3*, *PAX5*, *TBL1XR1*, *IGLL5* and *APC*) in previous studies [15,39,40,41,42]. The nonsynonymous somatic variants detected in *TBL1XR1* and *PAX5* were all regarded as major clonal variants, i.e., present in the majority of tumor cells. The nonsynonymous somatic variants detected in *ID3* and *IGLL5* consisted of both major and minor clonal variants. The two nonsynonymous somatic variants detected in *APC* were regarded as minor clonal variants, i.e., present in a subset of tumor cells (Appendix A).

### 3.6. Copy Number Alterations

CNA data were available for 14 out of 17 cases. CNAs were detected in eight cases (57%) (Figure 3), whereas no CNAs were identified in 6 other cases. In total, 18 CNAs were identified, with a median of two CNAs (range 1–5) per case. Two cases had a copy number gain in a similar region on chromosome 2 and two cases had a copy number gain in a similar region on chromosome 19. Two cases had a copy number loss in a similar region on chromosome 6. The case with the highest number of SNVs also had the highest number of CNAs. None of the recurrent mutations coincided with any of the identified CNAs.

## 4. Discussion

In this study on the mutational landscape of pSS-associated MALT lymphoma, an unexpectedly low number of SNVs and InDels, as well as a low number of CNAs, was observed in 14 out of 17 cases. The mutational load in the majority of the pSS-associated MALT lymphomas was <15 variants/case. This is much lower than the median number of variants observed in non-pSS MALT lymphoma of the thyroid and salivary gland (mean number of 111 mutations/case without matched germline DNA [15]) and is among the lowest number of variants reported across various cancers [43]. The read depths of our whole-exome sequencing data were high enough to enable the detection of nonsynonymous somatic variants with a VAF above 5% in the tumor samples, which should be sufficient to detect somatic variants even in cases with only 10% tumor cells.

The mutational landscape of pSS-associated MALT lymphomas displayed a very limited overlap with the mutational landscapes of SMZL, nodal marginal zone lymphoma and non-pSS MALT lymphoma [11,14,15,17,18,19,20,21,23,24]. Four genes recently described to be mutated in the non-pSS MALT lymphoma of the salivary gland, *TBL1XR1*, *CCR6*, *TNFAIP3* and *NOTCH2*, were also mutated in a subset of our pSS-MALT lymphomas [15,24]. The nonsynonymous variants detected in *TBL1XR1* and *CCR6* were based on the high VAF considered to be present in the majority of the tumor cells. *NOTCH2*, a gene mutated in >25% of SMZL [14,18,19], was mutated in one pSS-associated MALT case. In addition, variants were detected in *ID3* and *IGLL5*, two genes recurrently mutated in Burkitt lymphoma [39,40]. These two genes were mutated in the two cases that relapsed outside of the salivary gland and carried a relatively high mutational load. Finally, *PAX5* has been associated with aggressive B-cell non-Hodgkin’s lymphoma and *APC* with adult T-cell leukemia/lymphoma [41,42].

Similar to the low mutational load, a limited number of CNAs was observed in the pSS-associated MALT lymphoma cases. In contrast, in non-pSS MALT lymphomas of the salivary gland trisomies of chromosome 3 and 18 have been detected in 12% and 36% of cases, respectively [16]. Translocations involving the *MALT1* gene, observed in 15–40% of the non-pSS MALT lymphomas irrespective of the site of presentations, with the t(11;18)(q21;21) being the most common reported translocation [14], were not observed in this study. Thus, our data indicate a limited overlap of the CNA profiles of pSS-associated with other non-pSS MALT lymphomas.

The low frequency of both SNVs and CNAs indicates an overall low degree of genomic instability in pSS-associated MALT lymphoma. Interestingly, three cases of pSS-associated MALT lymphoma were characterized by a higher mutational load. Unlike the other patients with a favorable course of pSS-associated MALT lymphoma after treatment, two of the three patients with a high tumor mutational burden were characterized by severe pSS activity and systemic relapses of their lymphoma. Of these two patients, one had a stage IV MALT lymphoma according to the Ann Abor classification at the time of diagnosis. The high mutational burden appears to be associated with a relapsing and remitting course, like that observed in non-pSS MALT lymphoma [14,44]. Thus, a high mutational burden may be associated with an advanced clinical stage of MALT lymphoma. The course of the disease in the third patient with higher mutational burden is unknown, as the patient was lost to follow-up. These observations can be translated to a clinical setting and potentially have value as biomarkers for identification of patients at risk of a more aggressive disease. However, as the sample size of our study is relatively small, confirmation in larger cohorts is required to establish this potential association.

The lack of typical lymphoma-associated variants in pSS-associated MALT lymphoma appears to contradict recent observations of common B-cell lymphoma-associated variants by single-cell RNA sequencing of circulating B-cells of four pSS patients that produce pathogenic cryoglobulins [4]. Mutations were identified in 48 genes recurrently mutated in non-Hodgkin’s lymphoma. Genes mutated in our study as well as in the latter publication were *ID3*, *KLHL6*, *IGLL5*, *NOTCH2*, *TBL1XR1*, *CREBBP*, *ARID1A*, *APC* and *MUC16*. The majority of the common variants (18 of 21 somatic variants) observed in our study were detected in the three pSS-associated MALT lymphomas with a relatively high mutational load. The limited overlap in somatic variants with the circulating B-cell profile is remarkable, since the same pool of cells, i.e., B-cells that express stereotypic RF, appears to be involved in both cryoglobulinemia and pSS-associated MALT lymphoma [10]. A potential explanation for this discrepancy is that the circulating B-cells and pSS-associated MALT lymphoma share a common ancestor but follow different evolutionary tracks. This shared ancestor with a stereotype immunoglobulin heavy-chain variable gene region could be shaped by the microenvironment, leading to two different outcomes. The hypothesis of different B-cell populations is clinically supported by the fact that cryoglobulins can be detected in pSS patients long before overt pSS-associated MALT lymphoma and can persist after successful treatment. The presence of multiple B-cell subclones is known for other lymphoproliferative diseases and plasma cell dyscrasias as well [45,46,47].

The low mutational burden and the lack of a clear lymphoma-related gene signature in the majority of the samples presented in this study indicate that pSS-associated MALT lymphomas are relatively genomically stable. Although the possibility that the tumor cells have a growth advantage under the influence of a few key mutations cannot be excluded, the low mutational burden suggests that pSS-associated MALT lymphomas might still be dependent on external stimulatory signals from the inflammatory state of the micro-environment induced by pSS disease activity.

## 5. Conclusions

To our knowledge, this study is the first to describe the genomic landscape of pSS-associated MALT lymphoma. The majority of pSS-associated MALT lymphomas are characterized by a relatively low mutational burden and a limited number of CNAs. Unlike non-pSS MALT lymphoma, the majority of pSS-associated MALT lymphomas do not show B-cell lymphoma-specific oncogenic aberrations. Therefore, localized pSS-associated MALT lymphomas are a distinct type of ENMZL, which are most likely still dependent on the stimulatory signals obtained from the increased inflammatory state of the micro-environment.

## Figures and Tables

**Figure 1 cancers-14-01010-f001:**
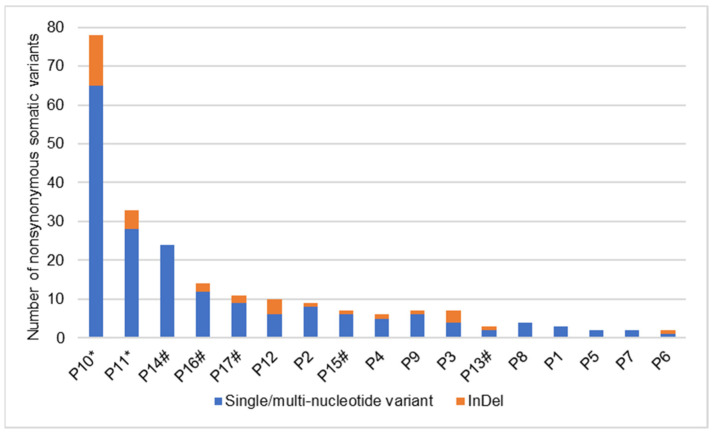
Number of nonsynonymous somatic variants detected in 17 cases with primary Sjogren’s syndrome related extranodal marginal zone lymphoma of mucosa-associated lymphoid tissue. The vertical axis represents the number of somatic variants. The horizontal axis represents the cases, showing 12 cases with a matched blood sample and 5 cases without a matched blood sample. (* relapsed outside of the salivary gland, # cases without a matched blood sample).

**Figure 2 cancers-14-01010-f002:**
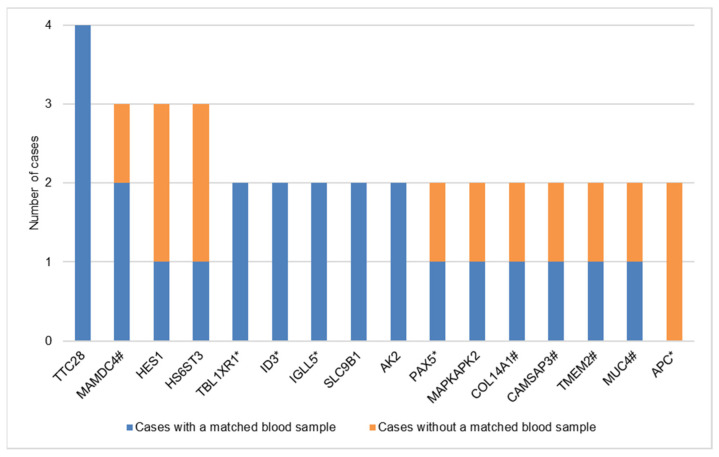
Recurrently mutated (>1 case) genes among 17 cases with primary Sjogren’s syndrome related extranodal marginal zone lymphoma of mucosa-associated lymphoid tissue. The vertical axis represents the number of cases. The horizontal axis represents the recurrently mutated genes. This figure shows 5 mutated genes known to be associated with lymphomagenesis (*ID3*, *TBL1XR1*, *PAX5*, *IGLL5* and *APC*) and 5 mutated genes involved in epithelial surface and/or extracellular matrix (*MAMDC4*, *COL14A1*, *CAMSAP3*, *TMEM2* and *MUC4*). (* genes associated with lymphomagenesis, # genes involved in the extracellular surface and/or extracellular matrix).

**Figure 3 cancers-14-01010-f003:**
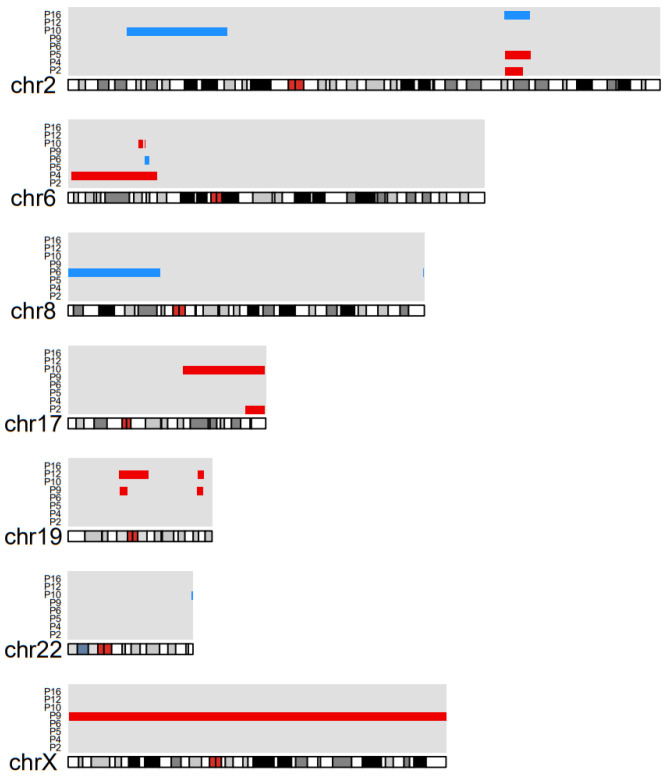
Copy number alterations detected in 8 out of 14 cases with primary Sjogren’s syndrome related extranodal marginal zone lymphoma of mucosa-associated lymphoid tissue. The vertical axis represents the cases with copy number alterations. The horizontal axis represents the location of the copy numbers detected in the displayed chromosomes. This figure shows a total of 18 copy number alterations, with a median of 2 copy number alterations (range 1–5) per case. (Blue: gain of copy number, red: loss of copy number).

## Data Availability

Sequencing data will be made available in the European Nucleotide Archive repository (accession number PRJEB48211 and ERP132550).

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
