# Peer review of "Low Mutational Burden of Extranodal Marginal Zone Lymphoma of Mucosa-Associated Lymphoid Tissue in Patients with Primary Sjogren’s Syndrome"

_cancers, 2022, doi:10.3390/cancers14041010_

Round 1
Reviewer 1 Report
Summary-The authors are investigating the genetic profile of MALT lymphoma in pSS associated patients. 17 fresh frozen or FFPE biopsies were exome sequenced for this study. The results indicate a low mutation rate leading to a conclusion that factors other than genetic instability is responsible for MALT lymphomas in pSS patients.
The study design and analysis methodology have certain issues that may have influenced the results.
- Its my understanding that the complete or the majority of the biopsy was used for DNA isolation which can create an imbalance of tumour to normal DNA ratio. Also, the germline DNA is contribute by PBMC which in case of immunological tumours can contain circulating tumour cells and are therefore, not appropriate match.
- The authors have set internal thresholds to differentiate the germline variants from the somatic variants. Use of variant callers such as Mutect, Strelka, SomaticSniper would eliminate the necessity manual read thresholds. Also, the current depth of sequencing though appropriate is insufficient for the use of two variant callers.
- Did the authors identify any somatic mutations that were present in the samples but not identified by the dbSNP database, the 1000 genome project or cosmic database?
- Classification of major and minor allele is unclear and requires further clarification.
- Figure 1 y-axis, is the number of cases an appropriate legend?
- Copy number alterations- did the gain or loss was observed to influence any significant genes with in the region of chromosome?
- Lastly the authors claim in discussion that the small number of somatic mutations means that genetic instability is not the probable cause of tumour formation in these patients. But the identified mutations are specifically in genes that enhance tumour growth and therefore, the few key mutations are sufficient for tumour formation? that is a hypermutations is not required for tumour growth in these patients? have authors considered this possibility?
Author Response
Please see the attachment (page 2-4).

Reviewer 2 Report
The manuscript: «Low mutational burden of extranodal marginal zone lymphoma of mucosa-associated lymphoid tissue in patients with primary Sjogren’s syndrome» aims to show a descriptive investigation about the genomic landscape of pSS-associated MALT lymphoma. This is an interesting study and will be an important source of information for the readership, however, several issues need to be addressed by the authors:
- The authors did not state that they used immunophenotypic findings to pSS-associated MALT lymphoma diagnosis.
- The authors did not explain why samples of 18 pSS patients with MALT lymphomas were collected. Was the selection of this number of patients according to the prevalence of the syndrome in that geographical region?
- The patients were different from each other in terms of stage. The discussion did not consider the patients with Stage IV who had a higher chance of having mutations and CNA. It is suggested that the authors should take into account not only the effect of higher stages on the occurrence of more mutations but also it is stated as a recommendation in the discussion.
- Although the authors, on and on, have mentioned that MALT lymphoma not associated with 38 pSS (non-pSS), they did not consider MALT stages. If previous studies have not regarded the stage of these patients, it is recommended to cite it in the discussion.
- It is recommended to compare WES of pss patients with and without MALT lymphoma ( as control) to remove non-lymphoma-related variants.
Author Response
Please see the attachment (page 5-6).

Reviewer 3 Report
Authors analyzed mutational landscape of patients with MALT lymphoma associated with Sjogren Syndrome.
The method is clear and results add more data to understand the heterogeneity of this lymphoma subtype.
The concern I have is purely the number of cases analyzed although I understand how hard it is to collect this specific condition.
You may want to add some limitation regarding the sample size, its hard to draw conclusions and discuss too much regarding the difference with other studies by single digit number of recurrent mutations.
Also, as 1 case was excluded from the analysis, I think we better exclude the patient from table and text (say 17 as total) to avoid confusion.
Author Response
Please see the attachment (page 7).

Round 2
Reviewer 1 Report
The authors have satisfactorily addressed the comments from previous submission.